# A-to-I RNA co-editing predicts clinical outcomes and is associated with immune cells infiltration in hepatocellular carcinoma

Juan Chen [1,4], Cheng-Hui Zhang[1,4], Tao Tao[1], Xian Zhang[1], Yan Lin[1], Fang-Bin Wang[1], Hui-Fang Liu [2] ✉ & Jian Liu [1,3] ✉

Aberrant RNA editing has emerged as a pivotal factor in the pathogenesis of hepatocellular carcinoma (HCC), but the impact of RNA co-editing within HCC remains underexplored. We used a multi-step algorithm to construct an RNA co-editing network in HCC, and found that HCC-related RNA editings are predominantly centralized within the network. Furthermore, five pairs of risk RNA co-editing events were significantly correlated with the overall survival in HCC. Based on presence of risk RNA co-editings resulted in the categorization of HCC patients into high-risk and low-risk groups. Disparities in immune cell infiltrations were observed between the two groups, with the high-risk group exhibiting a greater abundance of exhausted T cells. Additionally, seven genes associated with risk RNA co-editing pairs were identified, whose expression effectively differentiates HCC tumor samples from normal ones. Our research offers an innovative perspective on the etiology and potential therapeutics for HCC.

A-to-I RNA editing, catalyzed by the adenosine deaminase acting on RNA (ADAR) family of enzymes, represents the most prevalent form of RNA editing in human cells. This process involves the conversion of adenine nucleotides (Adenosine, A) to hypoxanthine nucleotides (Inosine, I), and due to inosine being interpreted as Guanosine (G) during translation, this type of editing is also known as A-to-G RNA editing[1]. Similar to DNA mutations, RNA editing plays a pivotal role in diversifying the transcriptome, thus providing innovative insights into the complexities of transcriptional regulation[2]. With the rapid development of high-throughput sequencing technology, researchers have found that there are a large number of A-to-I RNA editing sites in the human transcriptome. Comprehensive repositories, such as RADAR[3] and REDItools[4] provide a vast array of known A-to-I RNA editing sites.

Hepatocellular carcinoma (HCC) is a type of malignant tumor with high mortality and poor clinical prognosis worldwide. A growing number of studies have found that A-to-I RNA editing is strongly associated with the development of malignant tumors, including HCC. A-to-I RNA editing of the antizyme inhibitor 1 (AZIN1) gene in HCC resulted in a serine (S) to glycine (G) substitution at residue 367, promoting the malignant behavior of HCC[5]. Our previous investigation revealed extensive aberrant RNA editing events within tumor tissues from HCC patients, some of which can modulate cancer progression through effects on amino acid translation, alternative splicing, microRNA regulation and gene expression[6].

DNA mutations have long been considered a driving force behind the initiation and progression of tumors, and DNA co-mutations are prevalent in malignant tumors[7]. For example, the P53 and KRAS co-mutation has been reported to facilitate metastasis in pancreatic cancer[8], which can also be used as a prognostic marker for colorectal cancer and promote its metastasis[9,10]. In addition, certain DNA co-mutation pairs are intimately linked to the efficacy of tumor immunotherapy and resistance to chemotherapy[11,12]. Since RNA editing can mimic the genetic impact of DNA mutation, and may serve as a complementary mechanism to DNA mutations in the context of HCC risk genes in patients with HCC[6], it raises the question: whether RNA co-editing could serve as a clinical biomarker in HCC, and play a role in modulating the advancement of tumor progression?

[1]School of Food and Biological Engineering, Hefei University of Technology, Hefei 230009, China. [2]Department of Endocrinology, The Central Hospital of Wuhan, Tongji Medical College, Huazhong University of Science and Technology, Wuhan 430014 Hubei, China. [3]Engineering Research Center of Bio-process, Ministry of Education, Hefei University of Technology, Hefei 230009, China. [4]These authors contributed equally: Juan Chen, Cheng-Hui Zhang. ✉e-mail: Liu_HF91@163.com; liujian509@hfut.edu.cn

However, there is a paucity of research exploring the effect of RNA co-editing in malignancies, including HCC.

Tumor immune microenvironment plays an important role in recognizing and killing tumor cells, such as CD8+ T cell which is a specialized population of cytotoxic T lymphocytes for anti-cancer responses[13]. However, these CD8+ T cells may become dysfunctional or 'exhausted', characterized by impaired effector functions, increased expression of inhibitory receptors, such as programed cell death protein 1 (PD1), programmed death ligand 1 (PD-L1), T-cell immunoglobulin and mucin domain-containing protein 3 (TIM-3), contributing to tumor immune evasion[14]. A previous study reported that ADAR1-mediated hyper-editing of the cyclin I mRNA could generate major histocompatibility complex class (MHC)-presented epitopes recognized by the immune system in melanoma[15]. Considering the role of various DNA co-mutations in shaping responses to immunotherapy and chemotherapy[11,12], whether RNA co-editing events have an effect on immune cell infiltration and response to immunotherapy in HCC, deserves further analysis.

Biological networks can help us better understand and dissect the complexity of cancer pathogenesis, and serve as an essential platform for investigating key tumor molecules and carcinogenic mechanisms at a system level[16]. Through the construction of cancer-specific biological networks, various researchers have demonstrated the utility of network analysis in uncovering potential cancer prognostic biomarkers[17]. The principle of "guilt by association" underpins biological network analysis, enabling the identification of a broader spectrum of disease-related molecular aberrations[18].

In the present study, we integrated multi-omics data to systemically identify and construct an A-to-I RNA co-editing network in HCC utilizing a multi-step algorithm. The network topology was analyzed. Using the clinical information of tumor patients, RNA co-editing pairs associated with clinical prognosis were identified. Furthermore, HCC patients were divided into high-risk and low-risk groups, which showed different immune cell infiltration. This work will help us study the molecular mechanism of HCC from the perspective of RNA co-editing, and is of great significance for identifying new diagnostic and therapeutic targets for HCC.

## Results

### Characterization of A-to-I RNA co-editing interaction landscape in HCC

Here we proposed a multi-step algorithm to statistically infer A-to-I RNA co-editing interactions in HCC (Fig. 1). First, if a significant difference in gene expression is observed between edited and non-edited tumor samples, the gene is considered to be an editing-associated gene for the given A-to-I RNA editing event (Fig. 1a). Next, a four-step algorithm was performed to construct an A-to-I RNA co-editing network in HCC: first, potential A-to-I RNA co-editing pairs were identified based on the empirical observation that RNA co-editing occurs significantly more than expected. Then, we applied a permutation strategy to filter RNA co-editing pairs to identify candidate RNA co-editing events. According to the notion that RNA co-editing might synergistically regulate the same gene sets, we further filtered the candidate RNA co-editing pairs on the basis of sharing editing-associated genes. At last, we assembled all RNA co-editing pairs into an RNA co-editing interaction network, where nodes represent A-to-I RNA editing sites and edges represent their co-editing interactions (Fig. 1b).

In total, we obtained 86,822 interactions among 6415 A-to-I RNA editing sites (Fig. 2a). The degree distribution of the network followed a power law distribution, indicating that the RNA co-editing network is scale-free ($R^2 = 0.91$, Fig. 2b). By examining the genomic location of two RNA editing sites within each co-editing pair (Fig. 2c), we found that 14.44% of RNA co-editing pairs resided on same chromosomes (12,537/86,822, Fig. 2d), of which 42.97% located within the same gene regions. For the co-editing pairs on the same chromosome, more than half of the pairs were less than 10 Mb in terms of genome distance (57.53%, 7213/12,537, Fig. 2e), and 32.89% were less than 1 Kb (4123/12,537). Among the 12,537 RNA co-edited pairs on the same chromosome, involving 921 genes, of which 2457 co-editing pairs exhibited dysregulation of at least one genes between HCC

and adjacent normal samples, which were involved in 116 dysregulated genes (DEseq2, Benjamini-Hochberg correction, false discovery rate (FDR) < 0.05, with at least 2-fold changes, Supplementary data 1). These results require validation across multiple datasets.

### HCC-related RNA editing sites tend to be hub nodes and are enriched in network modules

Modularity is an important feature of a biological network. By using MCODE, we identified 21 modules. Top three modules with highest clustering scores were presented in Fig. 3a, and the genes involved in the module were significantly enriched in cancer associated pathways, such as apoptosis, cell cycle, RIG I like receptor signaling pathway and oxidative phosphorylation (Fig. 3b). For each biological network, another important property is the connectivity of a node, which reflects how often a node interacts with other nodes. Hub nodes with extremely high connectivity are always essential nodes in a network. Here we found that HCC-related RNA editing sites showed higher connectivity than other nodes (Fig. 3c, $p = 7.1e{-46}$, one-side Wilcoxon-Mann–Whitney test), and are significantly enriched in hub nodes and module nodes (Fig. 3d, e, $p = 2.5e{-43}$ and $p = 1.8e{-40}$, Fisher's exact test). Moreover, the mean shortest path length among HCC-related editing sites was found to be shorter than random selected nodes (Fig. 3f), revealing that HCC-related editing sites are closed connected with each other, which may be more informative regarding HCC initiation and progression.

### RNA co-editing pairs can predict patients' overall survival time

By using patients' clinical follow-up information, we investigated whether the presence of RNA co-editing pairs correlates with prognosis. Twelve RNA co-editing pairs were identified to be associated with overall survival (Supplementary Fig. 1). After adjusting the effect of single editing site, five co-editing pairs were identified to be risk events that can predict the clinical outcome of HCC patients (Fig. 4a, Supplementary Table 1). Additionally, even after adjusting the gender, ages, tumor stages, tumor grades and fetoprotein value of HCC patients, these five co-editing pairs still preserve the predictive significance (Supplementary Table 2). Moreover, patients with a greater number of risk RNA co-editing pairs presented a poorer prognosis, indicating a cumulative effect of risk co-editing pairs in HCC patients (Fig. 4b). Thus we classified HCC patients to be low-risk group and high-risk group depending on whether they had no risk co-editing pairs or at least one pair, respectively.

### HCC subgroups showed distinct immune microenvironments

As expected, the high-risk subgroup showed a poorer prognosis than low-risk group ($p = 1.38e{-09}$, log-rank test, Fig. 5a). Principal component analysis (PCA) indicated that samples could be roughly divided into three groups (Fig. 5b). Consistent with expectations, the high-risk group exhibited more advanced tumor histologic grades and pathologic stages (Two-sided fisher's exact test, both $p$ value < 0.05, Fig. 5c–e). Given the critical role of immune microenvironments in cancer initiation, progression and clinical outcome[19,20], we examined whether different HCC subgroups have different immune cell infiltration. The high-risk HCC group presented a higher percentage infiltration of dendritic cells, macrophages, neutrophils, B cells and CD4+ T cells compared to the low-risk group (two-sided Wilcoxon-Mann-Whitney test, FDR < 0.05, Fig. 5c, f).

Tumor-infiltrating CD8+ T cells play a key role in recognizing and eliminating tumor cells. However, we found high-risk HCC group had poorer survival with no significant differences in CD8+ T cell infiltration (Fig. 5c, f). As majority of the infiltrating CD8+ T cells become 'exhausted' and contribute to cancer immune evasion[13,14], we examined the expression levels of the marker genes in exhausted T cells[13]. It is suggested that the high-risk group showed higher expression of the marker genes in exhausted T cells, including *TCF7*, *PDCD1*, *CTLA4*, *TIGIT*, *IFNG*, *LAG3*, *TOX*, *TGFB1*, *TNF*, *IL10* and *HAVCR2* (two-sided Wilcoxon–Mann–Whitney test, $p < 0.05$, Fig. 6a), which may partly answer why high-risk subgroup had poorer clinical outcome. MHC-I is another key component in antigen

**Article**

## a Identification of editing-associated genes for each A-to-I RNA editing site.

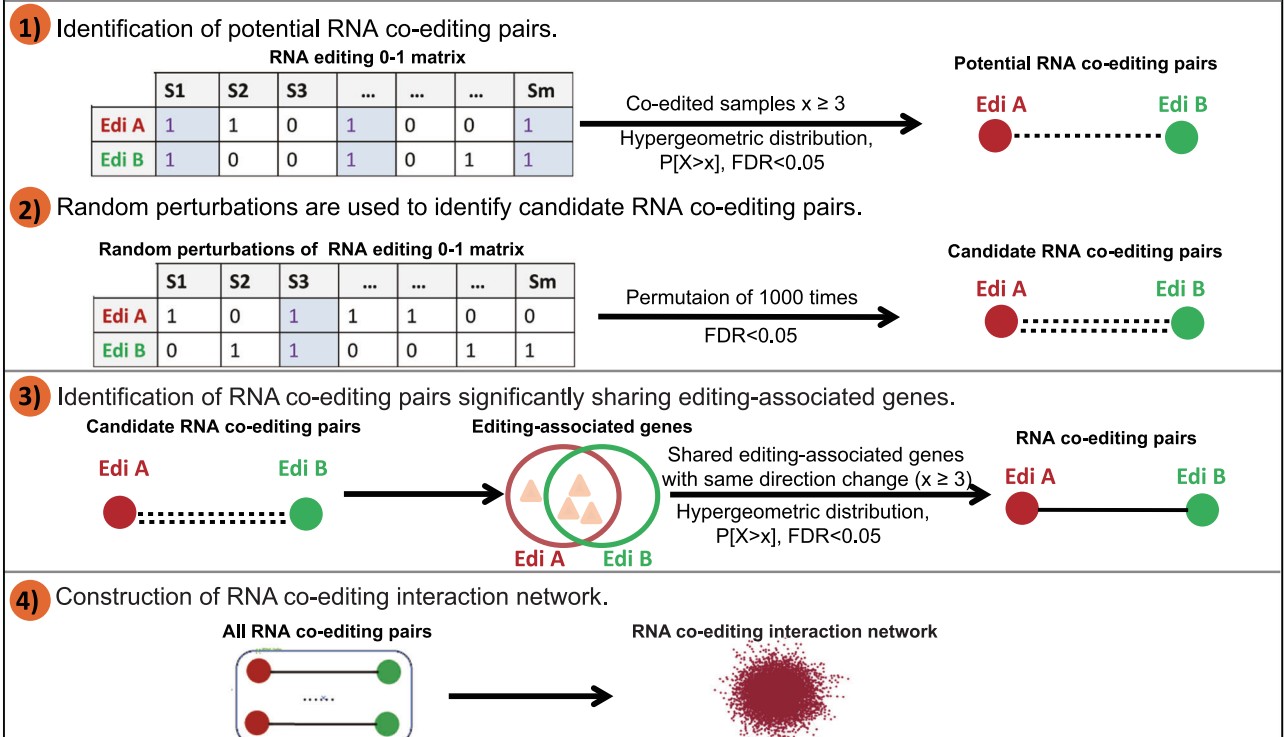

**Fig. 1 | Schematic diagram of the RNA co-editing pairs in hepatocellular carcinoma (HCC) and the workflow. a** Identification of candidate editing-associated gene sets for each A-to-I RNA editing site. **b** Construction of the A-to-I RNA co-editing network in HCC.

presentation, which presents peptide antigens to the cell surface for recognition by specific CD8+ T cells[21], improving the response of immune checkpoint blockade (ICB), such as anti-PD-1, anti-PD-L1, and anti-CTLA-4[22]. The mRNA expression levels of genes encoding the MHC-I components, including *HLA-A/B/C* and *B2M*, were investigated. As expected, high-risk group presented lower expression of *HLA-B*, *HLA-C* and *B2M* (two-sided Wilcoxon–Mann–Whitney test, $p < 0.05$, Fig. 6b). However, there was no significant difference in the expression of PD-L1 (encoded by *CD274*) between the two HCC subgroups.

**A seven-gene signature can predict the clinical status of samples**
To elucidate the mechanism by which risk RNA co-editing pairs influence HCC progression, we compared the gene expression profiles among normal, high-risk group and low-risk groups. This analysis lead to the identification of 418 shared differentially expressed genes, including 249 upregulated and 169 down-regulated genes using R package "Deseq2" (FDR < 0.05 & fold change >2, Fig. 7a). These differential genes were

significantly enriched in cancer-associated pathways, such as the cell cycle and P53 signaling pathway (hypergeometric test with Benjamini-Hochberg adjusted FDR < 0.05, Fig. 7b). Additionally, we intersected the editing-associated genes for each risk RNA co-editing pair, and obtained co-editing-associated genes for each pair. Among these 418 dysregulated genes, seven genes were also co-editing-associated genes shared by at least 3 risk RNA co-editing pairs, including *CA9*, *FRAS1*, *IGF2BP3*, *PPFIA4*, *PTGES3L*, *RCOR2* and *TPH1* (Fig. 7c). Globally, the expression levels of these seven genes were significantly elevated in HCC samples with RNA co-editing compared to those without (Supplementary Fig. 2, two-sided Wilcoxon-Mann-Whitney test, Benjamini-Hochberg adjusted FDR < 0.05). These genes can be used to distinguish tumors from normal samples with high sensitivity and specificity (Fig. 7d, logistic regression model, area under curve (AUC) = 0.93). The forecast performance of the model was validated in other four independent datasets from Gene Expression Omnibus (GEO) database (Fig. 7d, AUC = 0.97, 1, 0.93 and 0.85 for GSE65485, GSE169289, GSE77314, GSE164359, respectively). Notably, the GSE164359 dataset included

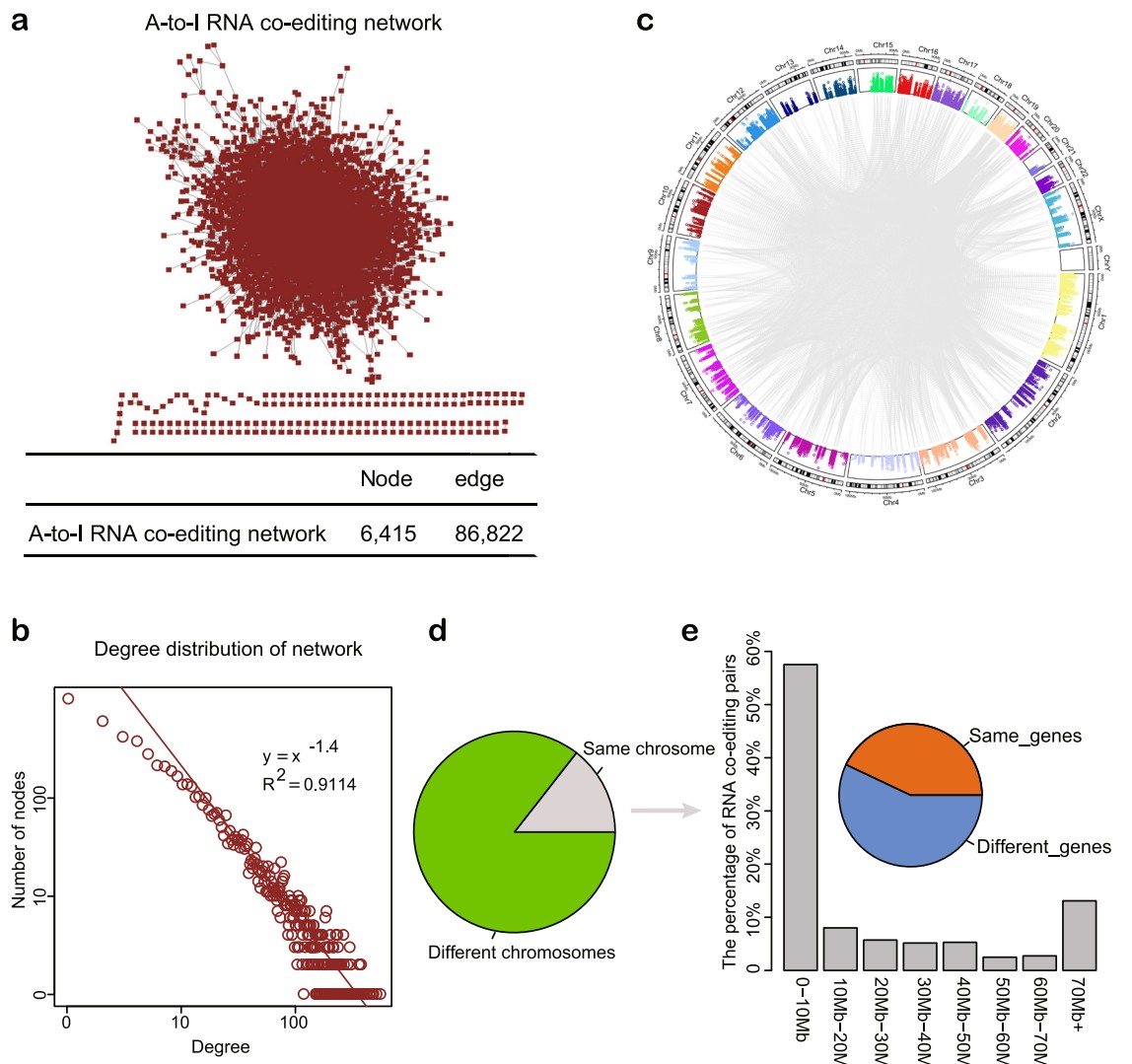

**Fig. 2 | The RNA co-editing network. a** Node and edges of network. **b** Degree distribution of the network. **c** Genomic distribution of RNA co-editing pairs. Each region along the circle represents one human chromosome, and edge denotes a co-editing interaction. **d** Chromosomes of RNA co-editing pairs located in. **e** The genomic distance of RNA co-editing pairs in the same chromosome.

samples from primary HCC and recurrent HCC, along with their adjacent normal liver samples. When we assessed the performance of the model respective in patients with primary and recurrent tumor, the predictive model can perfectly distinguish tumors from normal samples (Fig. 7d, AUC = 1 in both primary and recurrent HCC patients in GSE164359). Together, we hypothesis that these risk RNA co-editing pairs may lead to the expression disturbance of these seven genes, contributing to the progression of HCC.

## Discussion

Similar to DNA mutations, RNA editing contributes to transcript diversity[2], and previous research has focused on the involvement of DNA co-alterations in malignant tumor processes[8], there is a paucity of studies addressing RNA co-editing events in tumors and their potential involvement in the development of malignancies. We used a multi-step algorithm to construct an A-to-I RNA co-editing network specific to HCC, and identified 5 pairs of risk RNA co-editing that impact the overall survival of HCC patients. Based on risk RNA co-editing events, HCC samples could be divided into high-risk group and low-risk groups, which exhibited differences in immune cell infiltration, and our study provided an innovative perspective for investigating the carcinogenic mechanisms and therapeutic targets of HCC.

Given that A-to-I RNA editing is catalyzed by ADAR enzymes (ADAR1, ADAR2 and ADAR3)[1], we postulated whether RNA co-editing events were more prevalent in samples with elevated expression of these enzymes. Since ADAR3 is primarily expressed in the brain and is considered to be an enzymatically inactive deaminase[23], we focused on the expression of ADAR1 (encoded by ADAR in human) and ADAR2 (encoded by ADARB1 in human). We found that 38.86% RNA co-editing pairs were associated with higher *ADAR1* mRNA expression in co-edited samples, while 4.96% were associated with higher *ADAR2* expression (two-sided Wilcoxon-Mann-Whitney test, Benjamini-Hochberg adjusted, FDR < 0.05 Supplementary Data 2). Consequently, we surmised that the observed RNA co-editing pairs might depend on high expression levels of *ADAR1* and *ADAR2* to a certain extent. On the other hand, we considered the possibility that amplified genomic events could directly contribute to an increased frequency of co-editing pairs. To investigate this, we evaluated whether these RNA co-editing pairs tended to be co-amplified within genomic regions (hypergeometric test, Benjamini-Hochberg correction, FDR < 0.05). We discovered that 15.19% of RNA co-editing pairs exhibited co-amplification (13,184/86,822, Supplementary Fig. 3a), with the majority of these co-amplified RNA co-editing pairs being located on the same chromosome (85.54%, 11,277/13,184, Supplementary Fig. 3b). These results suggest that genomic co-amplification can lead to an increased occurrence of co-editing pairs.

**Fig. 3 | The network properties of HCC-related RNA editing sites. a** Top 3 network modules identified by MCODE. **b** KEGG pathway enrichment by top 3 network module genes (Hypergeometric test, Benjamini–Hochberg adjusted FDR < 0.05). **c** Node degree of HCC-related editing sites and other nodes (One-sided Wilcoxon–Mann–Whitney test). The boxplots are shown as median (line), interquartile range (box), and data range or 1.5× interquartile range (whisker). Percentage of hub nodes (**d**), and percentage of module nodes (**e**) of HCC-related editing sites and other nodes (Two-sided Fisher's exact test). **f** The mean shortest path of HCC-related editing sites and random nodes.

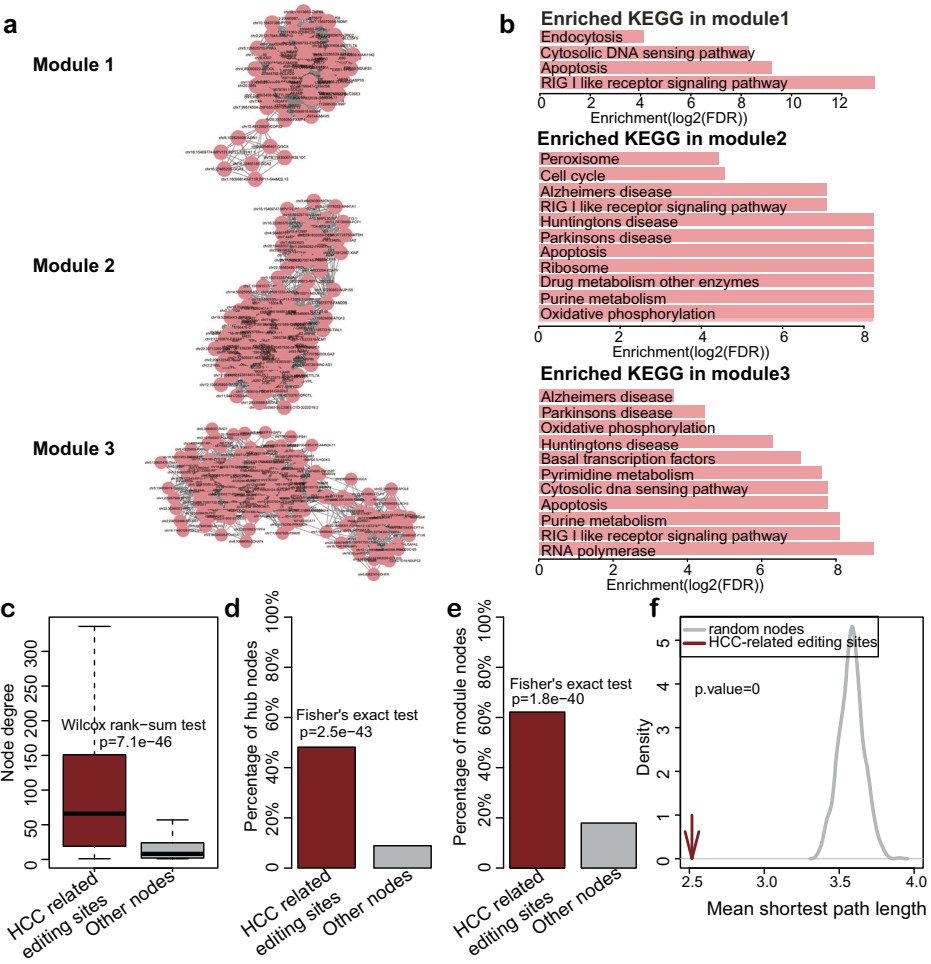

Although our results showed that the expression of genes are associated with A-to-I RNA co-editing event, we can't be sure this is a direct result of editing. However, previous studies reported that RNA editing may affect the expression level of a transcript directly or indirectly. For instance, APOL1 mRNA includes Alu elements at its 3'UTR that can form a double-stranded RNA susceptible to ADAR–mediated A-to-I RNA editing, potentially impacting gene expression[24]. Other studies reported that perturbations in RNA editing within the 3' UTR disrupt microRNA-mediated regulation of oncogenes and tumor suppressors[25]. Therefore, RNA editing may lead to essential changes in the overall expression level of a transcript, thus here we compared gene expression levels between edited and non-edited HCC samples to identify editing-associated genes. However, the impact of RNA editing on gene expression levels requires further experimental validation.

Debora Fumagalli and colleagues reported an increase in ADAR mRNA and protein expression in breast cancer cell lines following interferon treatment[26]. In our study, we further investigated the relationship between editing levels and the expression of interferon-related genes[27]. We focused on RNA editing sites within interferon-related genes that were edited in at least 25% of HCC samples. Among these, we identified three editing sites whose editing levels were significantly correlated with the genes' expression (Supplementary Fig. 4, Pearson correlation, Benjamini-Hochberg adjusted FDR < 0.05). Moreover, we explored the potential clinical prognostic value of interferon-related genes. However, after adjusting for age, gender, tumor pathologic stage, histologic grade, and alpha-fetoprotein levels, we found no significant correlation between the expression levels of interferon-related genes and the clinical prognosis of HCC patients (multivariate Cox regression analysis, Benjamini-Hochberg adjusted FDR > 0.05).

CD8+ T cell exhaustion is characterized by diminished effector functions and increased expression of inhibitory receptors, such as PD1 and CTLA4, accompanied by metabolic dysfunction and impaired proliferative capacity[28]. Therapeutic strategies targeting PD-1, PD-L1 and CTLA have elicited promising responses across various cancer types[29–31]. Here we found two HCC subgroups with distinct immune cells infiltration, especially high levels of exhausted T cells in the high-risk group, which may contribute to the poorer clinical outcome. Thus RNA co-editing maybe associated with tumor immune microenvironment, and deserves further investigation.

Given that HCC patients often face late diagnosis, poor prognosis and a lack of effective treatments, thus developing early diagnostic biomarkers is urgent nowadays[32]. Here we identified seven critical genes whose expression is associated with five risk RNA co-editing pairs. Next we validated the diagnostic efficacy of these seven genes across different datasets from HCC patients. These seven genes could be used as innovative diagnostic biomarkers for HCC, although further validation with more clinical datasets is necessary. Previous studies have also made a great advance in identifying diagnostic markers in HCC[33], such as a 3-protein marker consisting of HSP70, GPC3 and GS-(GLUL)[34], and another 3-gene signature comprising *GPC3*, *LYVE1* and *BIRC5*[35], have been shown to possess high predictive powers (Supplementary Fig. 5a, b). Notably, GPC3 is a common and promising marker reported to have high specificity[35,36]. When we combined our identified seven genes with *GPC3*, the performance was enhanced compared to the seven genes alone (Supplementary Fig. 5c).

Together, our findings highlight RNA co-editing is a valuable research view for investigating the mechanisms and identifying clinical biomarkers in HCC. Similar to DNA co-mutation, RNA co-editing may also affect the tumor immune microenvironment, and regulate the initiation and progression of HCC. This suggests that RNA co-editing has the potential to serve as a diagnostic biomarker and a therapeutic target for malignant tumors.

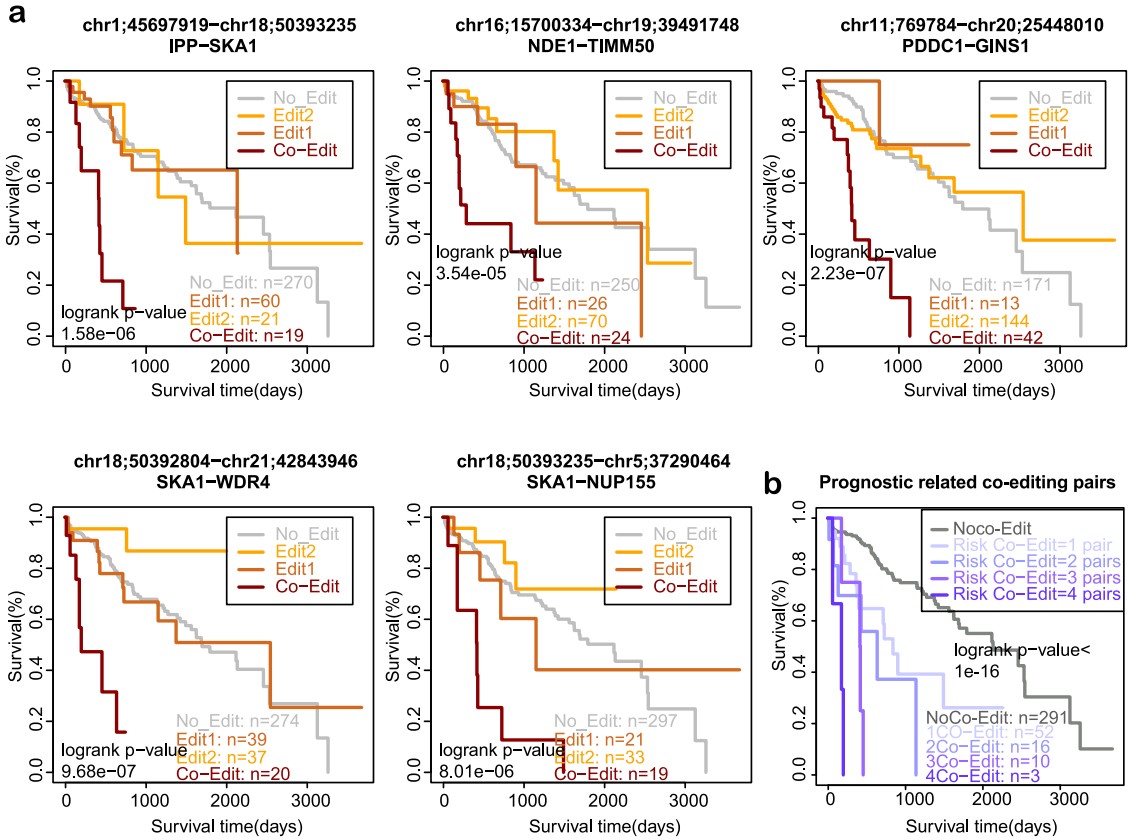

**Fig. 4 | The prognostic RNA co-editing pairs after adjusting the effect of single RNA editing sites. a** Five prognostic RNA co-editing pairs can predict the overall survival of HCC patients. **b** The more risk co-editing pairs a patient gets, the poorer the clinical outcome tends to be. The survival difference is calculated by the log-rank test.

## Methods
### Datasets

A-to-I RNA editing information of HCC and normal samples were originated from our previous study[6]. Briefly, RNA-seq BAM files of 373 HCC samples and 50 adjacent normal liver samples were downloaded from the database of Genotypes and Phenotypes (dbGaP)[37], and were converted to FASTQ format using BEDtools[38], and aligned to the human reference genome (GRCh38) by STAR[39]. Then the putative RNA editing sites were identified following the Best Practices recommendations for calling variants on the RNA-Seq data pipeline from the Genome Analysis Toolkit (GATK)[40]. At last, high-confidant RNA editing sites were obtained after removing DNA mutation sites and all known SNPs in dbSNP version 137[41] or the 1000 Genome Project[42]. To obtain high-confident A-to-I RNA editing sites, only editing sites which also annotated in RADAR database[3] were used for further analysis. The A-to-I RNA editing sites in RADAR database were converted from hg19 coordinates to hg38 coordinates by UCSC LiftOver tool.

The gene expression datasets, quantified as Fragments Per Kilobase of transcript per Million mapped reads (FPKM), along with the raw read counts generated by STAR software, and clinical information of HCC patients were downloaded from The Cancer Genome Atlas Program (TCGA). In addition to differential expression analysis using DEseq2, which used raw read count as a measure of expression level, other gene expression analyses were performed based on FPKM values. Only the genes expressed in at least 90% of samples were used for further analysis. The validation expression datasets for HCC samples were downloaded from Gene Expression Omnibus (GEO) database, including GSE65485[43], GSE169289[44], GSE77314[45] and GSE164359. The samples within GSE164359 can be divided into primary HCC and corresponding adjacent liver samples, as well as recurrent HCC and adjacent normal liver samples.

### Construction of the A-to-I RNA co-editing interaction network

Here we proposed a multi-step method to gradually identify A-to-I RNA co-editing pairs in HCC (Fig. 1). For each A-to-I RNA editing site, the RNA editing-associated genes were identified (Fig. 1a): initially, we compared gene expression levels between edited and non-edited HCC samples using a two-sided Wilcoxon-Mann-Whitney test, and the p value was subjected to a Benjamini–Hochberg correction for multiple tests. The genes with FDR < 0.05 and fold change >2 were selected as candidate editing-associated genes for a given RNA editing site. Next, to further avoid the effect of any other potential confounders, the logistic regression analysis was employed to adjust for age, gender, body mass index (BMI), pathologic stage, and histologic grade from edited and non-edited HCC samples (Benjamini-Hochberg adjusted FDR < 0.05). A four-step method was utilized to construct an A-to-I RNA co-editing network (Fig. 1b): first, for each pair of editing sites co-edited in at least three HCC samples, we identified the potential RNA co-editing pairs using a hypergeometric test (Benjamini-Hochberg adjusted FDR < 0.05). Subsequently, random perturbations were applied to further filter the potential RNA co-editing pairs and to identify the candidate RNA co-editing pairs. Permutated RNA editing matrices were produced by using the 'permatswap' function in the R package "vegan", which maintained the total number of editing samples for each site as well as the total number of editing events per sample. A total of 1,000 permutations were performed, and the proportion of permutations in which the observed co-editing exceeded the real situation was taken as an empirical p value (Benjamini-Hochberg adjusted FDR < 0.05), the co-editing pairs with permutated FDR < 0.05 were selected as candidate A-to-I RNA co-editing pairs. Next, for each candidate RNA co-editing pairs, hypergeometric distribution was used to identify A-to-I RNA co-editing pairs that significantly shared the editing-associated genes (with consistent direction of gene expression change, Benjamini-Hochberg adjusted FDR < 0.05). Finally, an A-to-I RNA co-editing network was constructed in HCC by assembling all

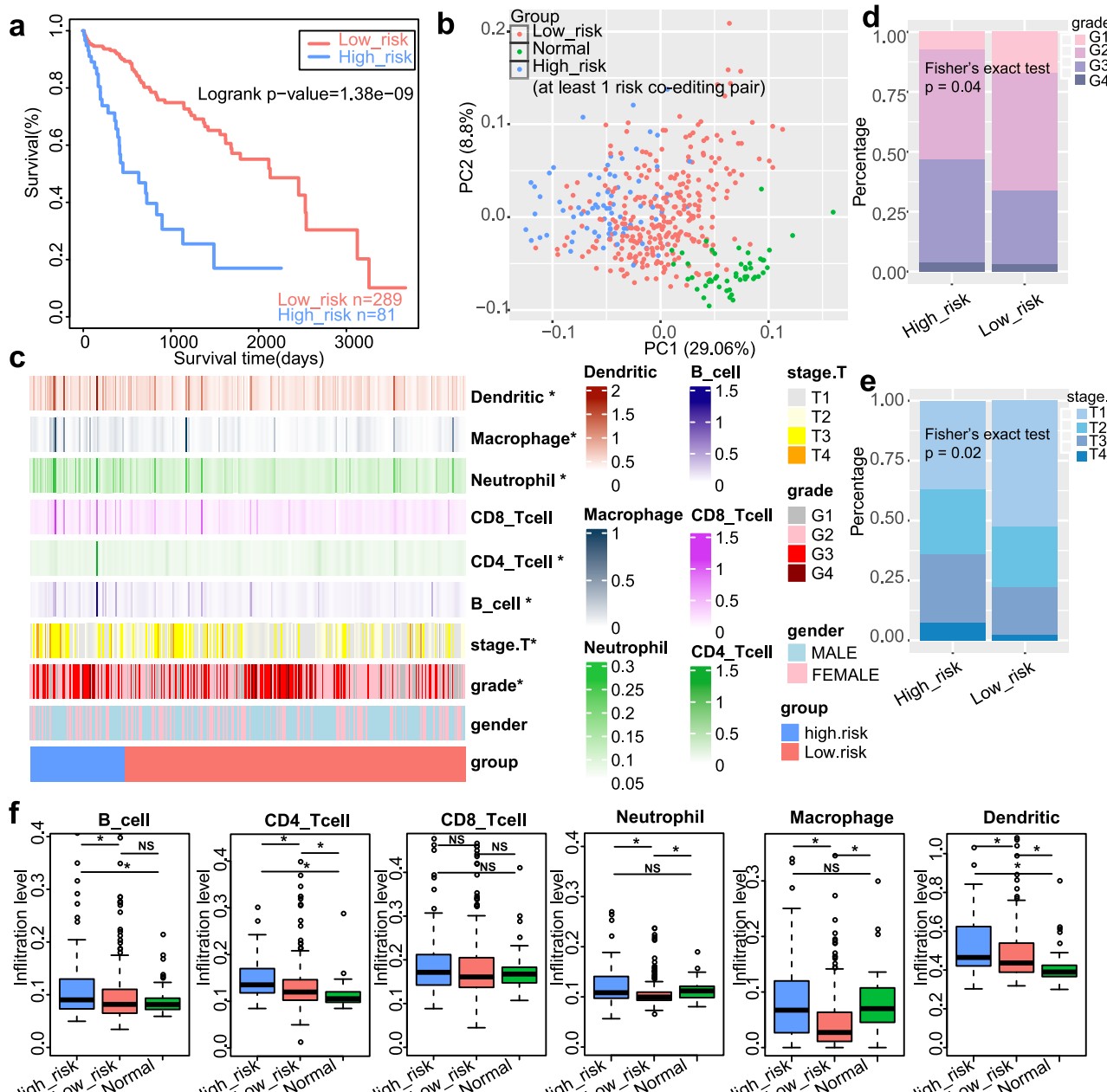

**Fig. 5 | The clinical characteristics between the high-risk group and low-risk group. a** Kaplan–Meier plots for the survival of HCC patients, which were separated into two groups based on the number of risk co-editing pairs that the patient carried: none and at least one pair, represents low-risk group and high-risk group, respectively. The survival difference is calculated by log-rank test. **b** Principal component analysis (PCA) for normal sample, low-risk group and high-risk group. 'PC' stands for principal component. **c** Heatmap of immune cell infiltration, age, gender, tumor stage, and tumor grade between low-risk group and high-risk group of HCC patients (Two-sided Wilcoxon–Mann–Whitney test. *FDR < 0.05). Tumor grades (**d**) and tumor stages (**e**) between the low-risk group and high-risk group (Two-sided Fisher's exact test). **f** Immune cell infiltration among normal samples, low-risk group and high-risk group (from TIMER database, two-sided Wilcoxon-Mann–Whitney test). The boxplots are shown as median (line), interquartile range (box), and data range or 1.5× interquartile range (whisker).

RNA co-editing pairs identified above. A node represents an A-to-I editing site, and two nodes are connected if they exhibit significant co-occurrence in HCC samples.

## Topological measurements of A-to-I RNA co-editing network

Several topological features of A-to-I RNA co-editing network were analyzed using the R package "igraph". The connectivity degree of a node was defined as the total number of edges connecting the node, and we evaluated whether the network's degree distribution satisfied a power law model. According to previous studies, top 10% of the nodes with the highest connectivity within a network were defined as hub nodes. The Molecular

Complex Detection Algorithm (MCODE) implemented in Cytoscape was utilized to identify network modules (Degree cutoff = 3, Node score cutoff = 0.2, k-core = 3, Max. Depth = 100).

HCC-related RNA editing sites were obtained from our previous study[6], including HCC gain, HCC loss, and dysregulated editing (dys-edit) sites (Supplementary Data 3). Briefly, HCC gain or HCC loss editing sites were identified by comparing the frequency of a particular RNA editing event in HCC and normal samples (Fisher's exact test with Benjamini–Hochberg correction, FDR < 0.05), with the additional criterion that these sites were edited in less than 5% of either normal or cancer samples, respectively. Dys-edit sites were detected by assessing the editing

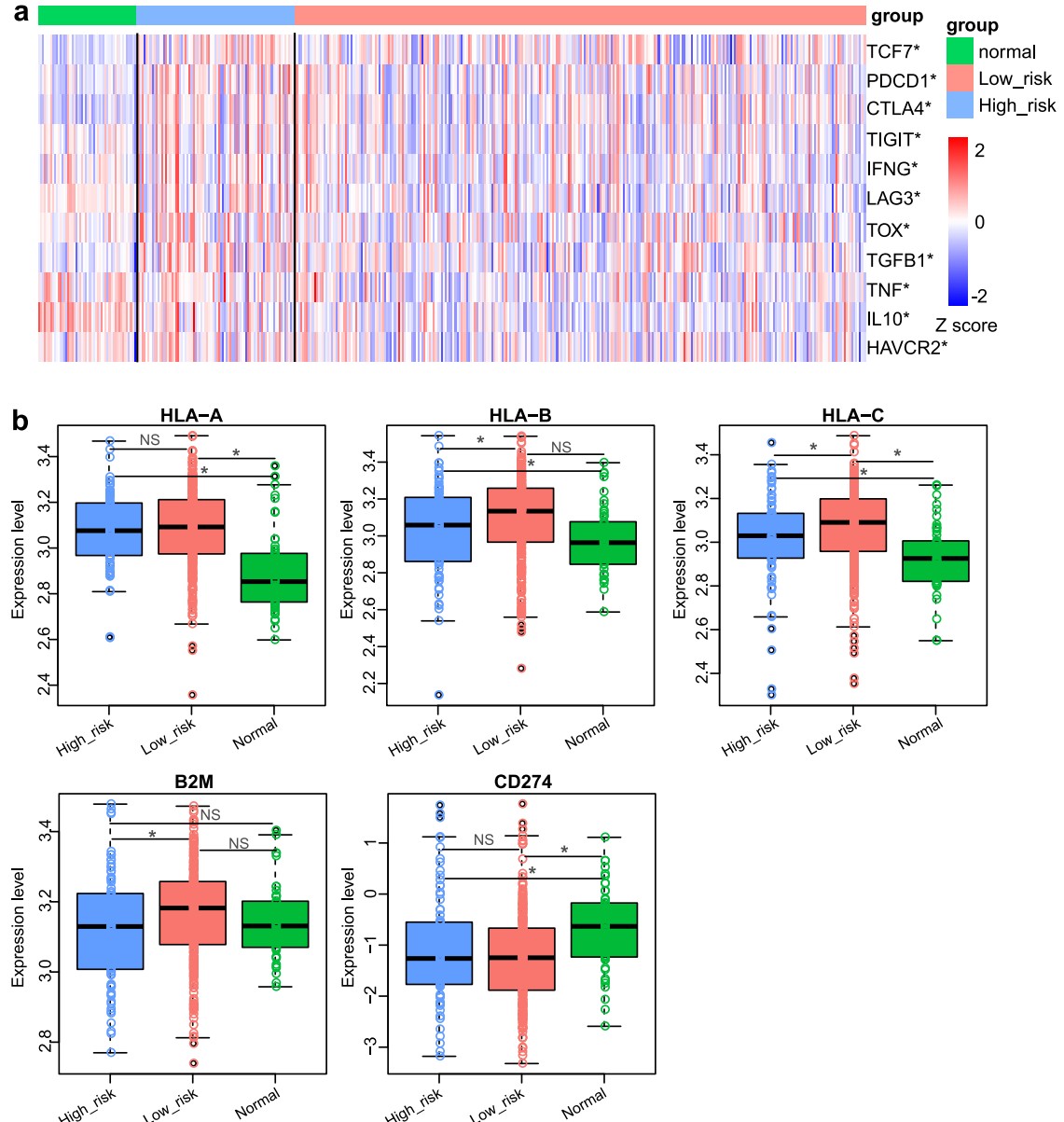

**Fig. 6 | T-cell exhaustion and MHC-I expression analysis between the low-risk group and high-risk group of HCC. a** The expression of T-cell exhaustion related markers between the low-risk group and high-risk group (Two-sided Wilcoxon–Mann–Whitney test. * p < 0.05). **b** MHC-I expression between the low-risk group and high-risk group (Two-sided Wilcoxon–Mann–Whitney test. * p < 0.05). The boxplots are shown as median (line), interquartile range (box), and data range or 1.5× interquartile range (whisker).

levels in 50 HCC tumor samples against their adjacent normal tissue counterparts, involving a two-step process: (i) a paired Student's t-test with Benjamini–Hochberg correction (adjusted FDR < 0.2 and p < 0.01); (ii) exhibiting an editing level change of more than 0.25 in at least two pairs of cancer and corresponding normal samples. Ultimately, we identified 242 HCC-related A-to-I RNA editing sites, 193 of which were present in the RNA co-editing network. We used a one-sided Wilcoxon–Mann–Whitney test to compare the connectivity differences between HCC-related editing sites and other nodes within the network. A two-sided Fisher's exact test was used to assess the enrichment of HCC-related editing sites in hub nodes and module nodes. The mean shortest path length was measured by using 'shortest.paths' function in the R package "igraph", including HCC-related editing nodes and other nodes randomly selected from network. A total of 1000 permutations were performed, with the proportion of permutations yielding a mean shortest path length lower than that observed in the actual data serving as the empirical p value.

## Prognosis associated analysis

For each A-to-I RNA co-editing site identified in HCC patients, we stratified patients into two groups: the co-edited group and the non-co-edited group. To evaluate the impact of RNA co-editing sites on patient survival, cox proportional hazard models were used for univariate survival analysis. This initial assessment was followed by multivariate survival analysis to adjust the effects of individual RNA editing events on overall survival. The log-rank test was used to assess if there were significant differences in overall survival among the co-edited group, the group with single site edit, and the group without any edits. The Kaplan–Meier plots were used to visualize the results. Consequently, five prognostic RNA co-editing pairs were identified as being associated with a poorer outcome, and were defined as risk RNA co-editing pairs. To determine whether these risk RNA co-editing pairs can provide additional predictive power, we performed a multivariate survival analysis by incorporating variables, such as age, gender, tumor grade, tumor

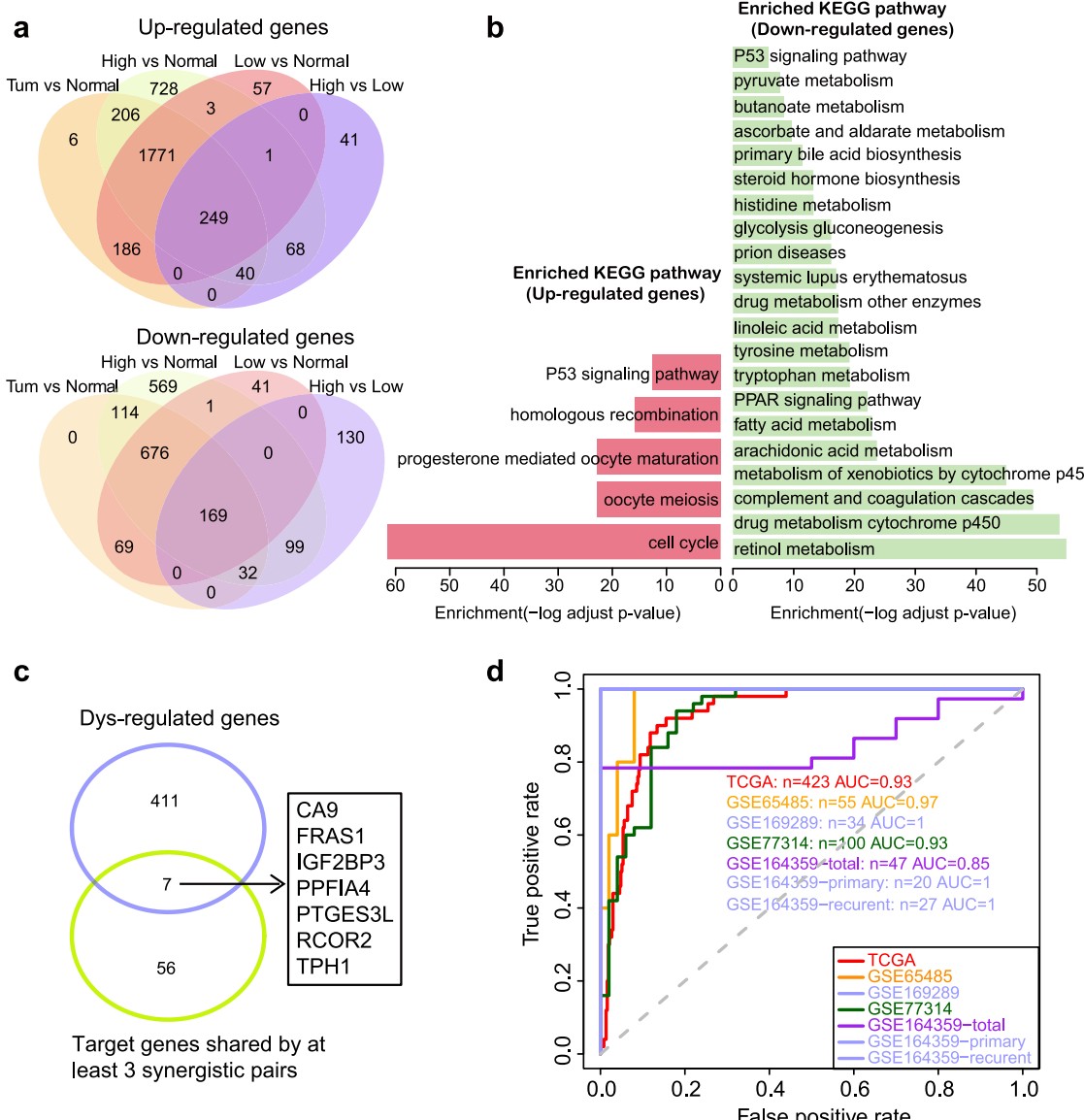

**Fig. 7 | Comparative analysis among normal sample, low-risk group and high-risk group of HCC. a** The intersection of dysregulated genes among normal sample, low-risk group and high-risk group. **b** KEGG pathway enrichment by shared 249 up-regulated genes and 149 down-regulated genes (Hypergeometric test, Benjamini–Hochberg adjusted FDR < 0.05). **c** Intersection of 418 dysregulated genes and co-editing-associated genes shared by at least 3 risk co-editing pairs. **d** ROC curves of seven genes to predict sample status in TCGA and four independent GEO cohorts (TCGA, GSE65485, GSE169289, GSE77314, GSE164359), and AUC values were listed.

stage, and fetoprotein levels of HCC patients, along with the presence or absence of RNA co-editing events.

For each HCC patient, we counted the number of risk RNA co-edited pairs that the patient carried, and divided the patients into different groups. The patients with a greater number of risk RNA co-edited pairs tended to have a poorer prognosis. Therefore, here we classified HCC patients into high-risk group and low-risk group based on whether they carried at least one risk RNA co-editing pair. The two-sided Fisher's exact test was used to assess the stage or grade difference between high-risk and low-risk groups (stage 1/2 vs stage 3/4, grade 1/2 vs grade 3/4).

### Immune cell infiltration associated analysis

The infiltration level of B cells, CD4 T cells, CD8 T cells, neutrophils, macrophages and dendritic cells were downloaded from TIMER database (http://cistrome.shinyapps.io/timer)[46]. A two-sided Wilcoxon–Mann–Whitney test was used to evaluate the differences in immune cell infiltration among the high-risk group, low-risk group and adjacent normal samples (Benjamini–Hochberg correction, FDR < 0.05).

Markers associated with exhausted T cells were originated from previous studies[13,22]. The expression differences of these markers between high-risk and low-risk groups were assessed using a two-sided Wilcoxon-Mann-Whitney test. The expression differences of genes encoding the MHC-I components, including *HLA-A/B/C* and *B2M*, as well as immune checkpoint genes such as *CD274* (encoding PD-L1) between high-risk and low-risk groups were also analyzed by the two-sided Wilcoxon-Mann-Whitney test.

### Dysregulated genes analysis

Raw STAR-counts of HCC samples and adjacent normal samples were downloaded from TCGA, and DESeq2 was used to identify differentially expressed genes among high-risk group, low-risk group and adjacent normal samples (FDR < 0.05, |log2(fold change)|>1). Gene sets from Kyoto Encyclopedia of Genes and Genomes (KEGG) were received from Molecular Signatures Database (MSigDB, V2023.1.Hs). Functional enrichment analysis of dysregulated genes was performed using a hypergeometric test (Benjamini–Hochberg correction, FDR < 0.05).

## Logistic regression model to predict sample types

We identified an intersection of 418 dysregulated genes and co-editing-associated genes of risk RNA co-editing pairs that were shared by at least three pairs, leading to the selection of seven critical genes for further analysis. The expression level of seven genes were used to construct logistic regression model to predict the status of samples. To assess how well a logistic regression model fits a dataset, the receiver operating characteristic (ROC) curve was created and AUC was calculated using the R package "ROCR". An AUC value closer to 1 indicates a better-fitting model.

## Copy number amplification analysis

The segmented copy number data for HCC samples were retrieved from the GDC portal, accessible at https://gdc.xenahubs.net. The log2 (copy-number/2) values larger than 0.30 was used to detect the amplifications in each HCC sample. For every pair of RNA co-editing sites, we determined the probability of observing no less than the actual number of samples that simultaneously co-amplified by hypergeometric test (Benjamini-Hochberg correction, FDR < 0.05).

## Statistics and reproducibility

The statistical significance of differences between groups was evaluated using Wilcoxon–Mann–Whitney test. The permutated RNA editing matrices were produced by using the 'permatswap' function in the R package "vegan". Fisher's exact test was used to determine whether or not there was a significant association between two categorical variables. The survival difference between groups was assessed by log-rank test. DESeq2 was used to identify differentially expressed genes among high-risk group, low-risk group and adjacent normal samples. A $p$ value less than 0.05 was considered statistically significant. The statistical analyses were performed by using R software version 4.2.2.

## Reporting summary

Further information on research design is available in the Nature Portfolio Reporting Summary linked to this article.

## Data availability

The gene expression profiles, clinical data were obtained from the TCGA Data Portal. The expression datasets of HCC were downloaded from GEO under the following accession numbers: GSE65485, GSE169289, GSE77314 and GSE164359. The KEGG annotation gene sets were downloaded from the MSigDB database (https://www.gsea-msigdb.org/gsea/msigdb/index.jsp). RNA editing sites of TCGA samples and HCC-related RNA editing sites were obtained from our previous study (Supplementary Data 3). The source data used to generate the main figures were provided in the Supplementary Data 3.

## Code availability

The code used in the work is available on GitHub (https://github.com/zchhui/RNA_co-editing/tree/HCC_2.0) with the identifier (https://doi.org/10.5281/zenodo.11218412). All software tools used in this study are open-source and freely available.

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

## Acknowledgements
We gratefully thank the TCGA Research Project for providing data for this work, and all employees of the National Institutes of Health (NIH) for managing the data. This work was supported by the National Natural Science Foundation of China [Grant Nos. 31801114 to J.C. and 32070757 to J.L.], Natural Science Foundation of Anhui Province [Grant No. 1908085QC98], and the Fundamental Research Funds for the Central Universities (JZ2022HGTB0246 to J.C.).

## Author contributions
J.C. designed study, performed research, analyzed data and wrote the paper, C.Z. performed research and analyzed data, T.T., X.Z., Y.L. and F.W. analyzed data, H.L. and J.L. designed study.

## Competing interests
The authors declare no competing interests.
