## [Transparent Peer Review file · Communications Biology]

A-to-I RNA co-editing predicts clinical outcomes and is associated with immune cells infiltration in hepatocellular carcinoma

Corresponding Author: Professor Jian Liu

Figures originally included in the author's rebuttal have been redacted from this file.

Version 0:

Reviewer comments:

Reviewer #1

(Remarks to the Author)

The work reports the analysis of co-editing patterns generated by the activity of ADAR enzymes and the correlations of these with outcome in hepatocellular carcinoma (HCC).

The major claims of the paper are that patterns of co-editing correlate with patient outcome. This appears to be the case in the available data.

The approach and data are novel as far as I am away but aspects of the methodology require clarification as they do not immediately make sense with the known biology of A-to-I editing. Specifically the method and text states "the target genes of A-to-I RNA editing sites were identified by comparing 228 gene expression level between edited and noedited HCC samples by Wilcoxon-Mann-Whitney test"- it is not clear to me what this means. If the authors have made the assumption than A-to-I editing leads to direct changes in gene expression (meaning overall transcript levels (TPM or similar) as assessed by RNAseq) then the authors need to provide evidence that this is actually the case and a direct result of editing of these transcripts. In most instances editing does not lead to significant changes in the overall expression level of a transcript (in human cells or mouse models). This needs clarification and further justification as it is central to the analysis method.

The conclusions report a correlation with outcome. It is not clear from the present work how this compares to other predictive markers in HCC if these exist. The authors should discuss this if they exist.

Other comments:

Editing by ADAR1 and 2 is dependent upon the RNA forming a double stranded structure. Can the correlation observed by determining likely structured RNAs expressed in the cells and ADR marking is just a surrogate of this rather than of any functional relevance?

Add patient numbers to figure 5A KM plot - it looks like only a small subset of the patient cohort would classify as high risk in this analysis.

Does the editing level correlate with interferon related gene expression? Does ISG expression overlap in terms of level of correlation with patient outcome? In breast cancer A-to-I editing and ADAR1 levels correlate with inflammation of genomic amplification of the locus (PMID: 26440892).

Reviewer #2

(Remarks to the Author)

This manuscript reports a study of RNA editing in hepatocellular carcinoma (HCC). Using an approach to identify co-editing sites, the authors identified a few pairs that correlated with patient survival. RNA co-edits also correlated, to certain degree, with immune cell infiltration or T cell exhaustion. Expression of genes related to these RNA co-edits could distinguish tumor and control samples. Although some of these results are interesting, the basic rationale behind the study is not well justified, the conclusions are often over-stated, and the text is not well-written.

The rationale behind the relevance of RNA co-editing in cancer is weak. Different from DNA mutations, RNA editing is catalyzed by ADAR enzymes. Why should co-editing occur in different genes? If they do matter, it is not justified to consider only pairs of RNA co-edits, rather than higher order relationships.

The method is not well defined. How are HCC related RNA editing sites identified? This is an important step of this work, so it warrants at least a brief description, not just referring to a previous paper. The method of identifying RNA co-editing is very confusing, and need more details, and improved writing.

Figure 5B: the low risk and high risk groups are not separated, the statement here is over strong (Line 136). The data do not support their conclusion.

Figure 5C: please show box plots or other summary plots to visualize overall differences between groups. Also, multiple testing correction is needed to determine a significance cutoff of p values.

Page 7: the concept of target genes of RNA co-edits is without any foundation. Such editing sites do not have a known function, those genes at best are somewhat correlated with those editing events. These are associations, not functional relationships.

Line 196: this is an overstatement. No evidence was shown in this paper that RNA co-editing may regulate immune microenvironment. Such causality was not established by their data.

Line 226-228: gene expression analysis needs to take into account potential confounders, such as age, gender and other variables.

There are many grammar mistakes throughout the manuscript. Please check carefully and correct them.

Writing: introduction section had highly similar contents repeated twice. Shorten it to be more succinct.

Reviewer #3

(Remarks to the Author)

In this study, the authors used bioinformatic approaches to explore whether HCC-related RNA editing could be used to predict the risk, survival and immune responses of HCC tumors. A multi-step algorithm has been developed to identify RNA co-editing pairs, which then were used to stratify patients into different risk groups. Based on the risk stratification, patient survival and immune responses have been analyzed and showed a significant correlation with RNA co-editing pairs and their related signaling pathways. Overall, the study provides new information of RNA editing in HCC biological changes and clinical outcomes in HCC tumors. There several questions regarding the data interpretation and analysis design:

1. In this analysis, 'if the expression of a gene is affected by an A-to-I RNA editing, that gene is considered to be a target gene for the given A-to-I RNA editing' is used as a parameter to define RNA editing genes. Please clarify in the text that whether the affected expression is defined by increase or decrease of gene expression. It remains unclear to what degree the authors might find RNA co-editing pairs may lead to different directions of gene expression change and how this situation was processed during the analysis.
2. In Figure 2C, the authors found that 14.36% of RNA co-editing pairs resided on same chromosomes. In tumor cells, a whole chromosome and/or certain regions of chromosomes are amplified. Cancer cells are not diploid cells. Is there a relationship between the number of RNA co-editing pairs and the amplification of chromosomal numbers/regions, which may lead to the enrichment of co-editing events in the same chromosomal or the same gene regions? Does the genomic amplification or increased chromosomal number (chromosomal instability) affect the identification of co-editing pairs since these amplified genomic events may directly lead to an increased occurrence of co-editing pairs? The genomic data (chromosome information, amplification status) of these co-editing pairs need to be assessed and discussed to exclude these possibilities in Figure 2D and 2E (same chromosome/gene or different chromosome/gene are very vague).
3. In addition, Figure 2E reported the gene location of RNA-editing. What are the targeting genes and what are the expression levels of these genes? These information could be reported to improve the clarity of analysis and also the interpretation of RNA-editing-associated changes.
4. In Figure 7, the up-regulated genes and down-regulated genes were identified from normal, low and high-risk groups. It remains to be unclear whether there is a correlation between these genes (Figure 7C) and the genes identified from Figure 2E, and to what degree these genes in Figure 7 could be affected or associated with RNA-editing activities.
5. In the line 200, the authors stated that 'we identified eight critical genes regulated by five risk co-editing pairs'. The data presented in Figure 7 is basically correlative analysis. These data do not provide causative relationship to support that the 8 genes are regulated by five risk co-editing. 'Associated' is a more accurate statement rather than 'regulated'.
6. In Figure 7D, multiple datasets were used to validate the prediction of genes identified from Figure 7C. However, can the authors validate their Figure 2D and Figure 2E results in multiple datasets? It is not clear whether repeatable or consistent RNA co-editing events can be identified from different datasets.
7. The language could be improved by avoiding some small grammar mistakes: e.g. line 57, 'Is RNA co-editing can be a clinical biomarker in HCC?'

Author Rebuttal letter:

Response to Reviewers' comments

COMMSBIO-23-1633 (A-to-I RNA co-editing predicts clinical outcomes and is associated with immune cells infiltration in hepatocellular carcinoma)

Reviewer #1 (Remarks to the Author):

The work reports the analysis of co-editing patterns generated by the activity of ADAR enzymes and the correlations of these with outcome in hepatocellular carcinoma (HCC). The major claims of the paper are that patterns of co-editing correlate with patient outcome. This appears to be the case in the available data.

1) The approach and data are novel as far as I am away but aspects of the methodology require clarification as they do not immediately make sense with the known biology of A-to-I editing. Specifically the method and text states "the target genes of A-to-I RNA editing sites were identified by comparing 228 gene expression level between edited and noedited HCC samples by Wilcoxon-Mann-Whitney test"- it is not clear to me what this means. If the authors have made the assumption than A-to-I editing leads to direct changes in gene expression (meaning overall transcript levels (TPM or similar) as assessed by RNAseq) then the authors need to provide evidence that this is actually the case and a direct result of editing of these transcripts. In most instances editing does not lead to significant changes in the overall expression level of a transcript (in human cells or mouse models). This needs clarification and further justification as it is central to the analysis method.

Reply: Thank the reviewer for the comment and suggestion. We agree with the reviewer's point and apologize for our unclear description. We changed the sentences to be "For each A-to-I RNA editing site, the RNA editing-associated genes were identified (Fig. 1A): initially, we compared gene expression levels between edited and non-edited HCC samples using a two-sided Wilcoxon-Mann-Whitney test, and the p value was subjected to a Benjamini-Hochberg correction for multiple tests. The genes with FDR < 0.05 and fold change > 2 were selected as candidate editing-associated genes for a given RNA editing site". Although our results showed that the expression of genes are associated with A-to-I RNA editing event, we can't be sure this is a direct result of editing. However, previous studies reported that RNA editing may affect the expression level of a transcript directly or indirectly. For instance, APOL1 mRNA includes Alu elements at its 3'UTR that can form a double-stranded RNA susceptible to ADAR-mediated A-to-I RNA editing, potentially impacting gene expression (Riella CV, et al., Proc. Natl. Acad. Sci. U. S. A. 2022). Previous research reveals that perturbations in RNA editing within the 3' UTR disrupt microRNA-mediated regulation of oncogenes and tumor suppressors (Zhang L, et al., Sci. Rep. 2016). Therefore, RNA editing may lead to significant changes in the overall expression level of a transcript, thus we changed the "editing-target genes" to be "editing-associated genes" to be more rigorous. We also discussed this in our revised manuscript (line 217-226) as shown below.

Although our results showed that the expression of genes are associated with A-to-I RNA co-editing event, we can't be sure this is a direct result of editing. However, previous studies reported that RNA editing may affect the expression level of a transcript directly or indirectly. For instance, APOL1 mRNA includes Alu elements at its 3'UTR that can form a double-stranded RNA susceptible to ADAR-mediated A-to-I RNA editing, potentially impacting gene expression²⁴. Previous research reveals that perturbations in RNA editing within the 3' UTR disrupt microRNA-mediated regulation of oncogenes and tumor suppressors²⁵. Therefore, RNA editing may lead to significant changes in the overall expression level of a transcript, thus here we compared gene expression levels between edited and non-edited HCC samples to identify editing-associated genes. However, the impact of RNA editing on gene expression levels requires further experimental validation.

2) The conclusions report a correlation with outcome. It is not clear from the present work how this compares to other predictive markers in HCC if these exist. The authors should discuss this if they exist.

Reply: We thank the reviewer for the suggestion. We searched other predictive markers in HCC from previous studies, such as a 3-protein marker, including HSP70, GPC3 and GS-(GLUL) (Di Tommaso L, et al., J Hepatol., 2009), and another 3-gene signature comprising GPC3, LYVE1 and BIRC5 (Llovet JM, et al., Gastroenterology, 2006), which also show high predictive powers (Fig. S5A, B). Notably, GPC3 is a common marker among these, reported to have high specificity (An S, et al., Eur J Nucl Med Mol Imaging, 2022). Here, we integrated the seven genes here we identified with GPC3, which showed an enhanced performance compared to the seven genes alone (Fig S5C). Here we discussed this in our revised manuscript (line 249-254) as shown below.

Previous studies have also made a great advance in identifying diagnostic markers in HCC³³, such as a 3-protein marker consisting of HSP70, GPC3 and GS-(GLUL)³⁴, and another 3-gene signature comprising GPC3, LYVE1 and BIRC5³⁵, have been shown to possess high predictive powers (Fig. S5A, 5B). Notably, GPC3 is a common and promising marker reported to have high specificity^{35,36}. When we combined our identified seven genes with GPC3, the performance was enhanced compared

to the seven genes alone (Fig. S5C).

Other comments:

3) Editing by ADAR1 and 2 is dependent upon the RNA forming a double stranded structure.

Can the correlation observed by determining likely structured RNAs expressed in the cells and ADR marking is just a surrogate of this rather than of any functional relevance?

Reply: We appreciate the reviewer for this comment and suggestion. As A-to-I RNA editing is mediated by ADAR1 and ADAR2, we speculated whether RNA co-editing preferred to be occurred in samples with higher expression of the ADAR enzymes. It is revealed that 38.86% of the RNA co-editing samples had higher ADAR1 expression compared to nonco-edited samples, and 4.96% have higher ADAR2 expression. Therefore, we hypothesized that the observed RNA co-editing pairs relied on high levels of ADAR1 and ADAR2 expression to a certain extent, which we have discussed in the revised manuscript (line 200-209) as shown below.

Given that A-to-I RNA editing is catalyzed by ADAR enzymes (ADAR1, ADAR2 and ADAR3)¹, we postulated whether RNA co-editing events were more prevalent in samples with elevated expression of these enzymes. Since ADAR3 is primarily expressed in the brain and is considered to be an enzymatically inactive deaminase²³, we focused on the expression of ADAR1 (encoded by ADAR in human) and ADAR2 (encoded by ADARB1 in human). We found that 38.86% RNA co-editing pairs were associated with higher ADAR1 mRNA expression in co-edited samples, while 4.96% were associated with higher ADAR2 expression (two-sided Wilcoxon-Mann-Whitney test, Benjamini-Hochberg adjusted, FDR < 0.05 Supplementary table S4). Consequently, we surmised that the observed RNA co-editing pairs might depend on high expression levels of ADAR1 and ADAR2 to a certain extent.

4) Add patient numbers to figure 5A KM plot - it looks like only a small subset of the patient cohort would classify as high risk in this analysis.

Thank the reviewer for this suggestion and we have added patient numbers to figure 5A.

5) Does the editing level correlate with interferon related gene expression? Does ISG expression overlap in terms of level of correlation with patient outcome? In breast cancer A-to-I editing and ADAR1 levels correlate with inflammation of genomic amplification of the locus (PMID: 26440892).

Reply: Thank the reviewer for this suggestion. We downloaded type I interferon related genes from the literature. To measure whether the editing levels correlate with the expression levels of interferon related genes, we retained the RNA editing sites of interferon related genes which edited in at least 25% HCC samples, and found three editing sites whose editing levels are correlated with the genes' expression levels (Fig. S4, Pearson correlation, FDR<0.05, BH corrected). In addition, we investigated the clinical prognostic value of interferon related genes. After adjusting the age, gender, tumor stages, tumor grade and fetoprotein value, the expression level of interferon related genes were not significantly correlated with the clinical prognosis of HCC patients (multivariate Cox regression analysis, adjust p>0.05, data not shown). We have discussed this in our revised manuscript (line 227-236) as shown below.

Debora Fumagalli and colleagues reported an increase in ADAR mRNA and protein expression in breast cancer cell lines following interferon treatment²⁶. In our study, we further investigated the

relationship between editing levels and the expression of interferon-related genes. We focused on RNA editing sites within interferon-related genes that were edited in at least 25% of HCC samples. Among these, we identified three editing sites whose editing levels were significantly correlated with the genes' expression (Fig. S4, Pearson correlation, Benjamini-Hochberg adjusted FDR < 0.05). Moreover, we explored the potential clinical prognostic value of interferon-related genes. However, after adjusting for age, gender, tumor stage, tumor grade, and alpha-fetoprotein levels, we found no significant correlation between the expression levels of interferon-related genes and the clinical prognosis of HCC patients (multivariate Cox regression analysis, adjusted p>0.05).

Reviewer #2 (Remarks to the Author):

1) This manuscript reports a study of RNA editing in hepatocellular carcinoma (HCC). Using an approach to identify co-editing sites, the authors identified a few pairs that correlated with patient survival. RNA co-edits also correlated, to certain degree, with immune cell infiltration or T cell exhaustion. Expression of genes related to these RNA co-edits could distinguish tumor and control samples. Although some of these results are interesting, the basic rationale behind the study is not well justified, the conclusions are often over-stated, and the text is not well-written.

Reply: We thank the reviewer for the comment and suggestions. We agree with the reviewer's point and checked the conclusions carefully to avoid any over-stated descriptions. We feel sorry for our poor writings. We have tried our best to improve the manuscript and have modified some confusing sentences, making them concise and easy to read. The changes have been marked in red in the revised manuscript, and we hope it could be acceptable for you.

2) The rationale behind the relevance of RNA co-editing in cancer is weak. Different from DNA mutations, RNA editing is catalyzed by ADAR enzymes. Why should co-editing occur in different genes? If they do matter, it is not justified to consider only pairs of RNA co-edits, rather than higher order relationships.

Reply: Thank the reviewer for the suggestions. Please see the response to the point 3 raised by the reviewer 1.

3) The method is not well defined. How are HCC related RNA editing sites identified? This is an important step of this work, so it warrants at least a brief description, not just referring to a previous paper. The method of identifying RNA co-editing is very confusing, and need more details, and improved writing.

Reply: Thank the reviewer for the suggestions. Sorry for our unclear description and we have added a brief description about the identification of HCC related RNA editing sites (line 335-346), and detailed the method of identifying RNA co-editing pairs in the revised version (line 300-325).

Line 319-329:

HCC-related RNA editing sites were obtained from our previous study⁶, including HCC gain, HCC loss, and significant dysregulated editing (dys-edit) sites (Supplementary data). Briefly, HCC gain or HCC loss editing sites were identified by comparing the frequency of a particular RNA editing event in HCC and normal samples (Fisher's exact test with Benjamini-Hochberg correction, $FDR < 0.05$), with the additional criterion that these sites were edited in less than 5% of either normal or cancer samples, respectively. Dys-edit sites were detected by assessing the editing levels in 50 HCC tumor samples against their adjacent normal tissue counterparts, involving a two-step process: (i) a paired Student's t-test with Benjamini-Hochberg correction (adjusted $FDR < 0.2$ and $p < 0.01$); (ii) filtering for sites exhibiting an editing level change of more than 0.25 in at least two pairs of cancer and corresponding normal samples. Ultimately, we identified 242 HCC-related A-to-I RNA editing sites, 193 of which were present in the RNA co-editing network.

Line 285-309:

Here we proposed a multi-step method to gradually identify A-to-I RNA co-editing pairs in HCC (Fig. 1). For each A-to-I RNA editing site, the RNA editing-associated genes were identified (Fig. 1A): initially, we compared gene expression levels between edited and non-edited HCC samples using a two-sided Wilcoxon-Mann-Whitney test, and the p value was subjected to a Benjamini-Hochberg correction for multiple tests. The genes with $FDR < 0.05$ and fold change > 2 were selected as candidate editing-associated genes for a given RNA editing site. Next, to further avoid the effect of any other potential confounders, the logistic regression analysis was employed to adjust for age, gender, body mass index (BMI), tumor stage, and grade from edited and non-edited HCC samples (Benjamini-Hochberg adjusted $FDR < 0.05$). A four-step method was utilized to construct an A-to-I RNA co-editing network (Fig. 1B): first, for each pair of editing sites co-edited in at least three HCC samples, we identified the potential RNA co-editing pairs using a hypergeometric test (Benjamini-Hochberg adjusted $FDR < 0.05$). Subsequently, random perturbations were applied to further filter the potential RNA co-editing pairs and to identify the candidate RNA co-editing pairs. Permutated RNA editing matrices were produced by using the `apermat` function in the R package `avegan`, which maintained the total number of editing samples for each site as well as the total number of editing events per sample. A total of 1,000 permutations were performed, and the proportion of permutations in which the observed co-editing exceeded the real situation was taken as an empirical p value (Benjamini-Hochberg adjusted $FDR < 0.05$), the co-editing pairs with permutated $FDR < 0.05$ were selected as candidate A-to-I RNA co-editing pairs. Next, for each candidate RNA co-editing pairs, hypergeometric distribution was used to identify A-to-I RNA co-editing pairs that significantly shared the editing-associated genes (with consistent direction of gene expression change, Benjamini-Hochberg adjusted $FDR < 0.05$). Finally, an A-to-I RNA co-editing network was constructed in HCC by assembling all significant RNA co-editing pairs identified above. A node represents an A-to-I editing site, and two nodes are connected if they exhibit significant co-occurrence in HCC samples.

4) Figure 5B: the low risk and high risk groups are not separated, the statement here is over strong (Line 136). The data do not support their conclusion.

Reply: We agree with the reviewer's point. Sorry for our over-stated description and we have changed the state to "Principal component analysis (PCA) showed the samples could be roughly divided into three groups" in our revised manuscript (line 142-143) as shown below.

Principal component analysis (PCA) indicated that samples could be roughly divided into three groups (Figure 5B).

5) Figure 5C: please show box plots or other summary plots to visualize overall differences between groups. Also, multiple testing correction is needed to determine a significance cutoff of p values.

Reply: We thank the reviewer for this suggestion. We have showed bar plots and box plots to visualize overall differences (Fig. 5D-F). After multiple testing correction, it is revealed that high-risk HCC group presented higher percentage infiltration of dendritic cell, macrophage, neutrophil, B cell and CD4+ T cell compared to low-risk group (two-sided Wilcoxon-Mann-Whitney test, $FDR < 0.05$, Fig. 5C, 5F), with no significant differences in CD8+ T cell infiltration. We have modified this in our revised manuscript (line 151-152, line 360-363) as shown below.

Line 151-152:

However, we found high-risk HCC group had poorer survival with no significant differences in CD8+ T cell infiltration (Figure 5C, 5F).

Line 360-363:

A two-sided Wilcoxon-Mann-Whitney test was used to evaluate the differences in immune cell infiltration among the high-risk group, low-risk group and adjacent normal samples (Benjamini-Hochberg correction, $FDR < 0.05$).

6) Page 7: the concept of target genes of RNA co-edits is without any foundation. Such editing sites do not have a known function, those genes at best are somewhat correlated with those editing events. These are associations, not functional relationships.

Reply: We agree with the reviewer's point and thanks for your suggestion. We changed the concept of target genes to be "editing-associated genes" in our revised manuscript.

7) Line 196: this is an overstatement. No evidence was shown in this paper that RNA co-editing may regulate immune microenvironment. Such causality was not established by their data.

Reply: We agree with the reviewer's point and we apologize for the overstatements throughout the manuscript. In the revised manuscript, we have changed this description in the revised manuscript (line 242-243) as shown below.

Thus RNA co-editing maybe associated with tumor immune microenvironment, and deserves further investigation.

8) Line 226-228: gene expression analysis needs to take into account potential confounders, such as age, gender and other variables.

Reply: Thanks very much for the suggestion and we accepted the advice. In the revised manuscript, we performed the logistic regression to further filter the editing-associated genes by adjusting age, gender, BMI, tumor stage and tumor grade. For each candidate RNA co-editing pairs, we keep A-to-I RNA co-editing pairs which significantly sharing the editing-associated genes with same direction changes of expression by hypergeometric test (BH correction, $FDR < 0.05$). Finally, an A-to-I RNA co-editing network was re-constructed with 86,822 interactions among 6,415 A-to-I RNA editing sites. We re-analyzed the results based on the reconstructed network, and found that although our approach for constructing the network has become more rigorous, the subsequent results have remained largely unchanged. The original 5 pairs of RNA co-editing were still associated with the patient's clinical outcome, but the number of shared editing-associated genes was changed from eight to seven. We have modified corresponding results in our revised manuscript.

Line 290-293

Next, to further avoid the effect of any other potential confounders, the logistic regression analysis was employed to adjust for age, gender, body mass index (BMI), tumor stage, and grade from edited and non-edited HCC samples (Benjamini-Hochberg adjusted $FDR < 0.05$).

Line 99

In total, we obtained 86,822 interactions among 6,415 A-to-I RNA editing sites (Figure 2A).

9) There are many grammar mistakes throughout the manuscript. Please check carefully and correct them.

Reply: Thanks the reviewer for this suggestion. We have tried our best to improve the manuscript and have corrected and modified some mistakes or confusing sentences, making them concise and easy to read. In addition, the modified manuscript has been reviewed and proofread by several professors to avoid any potential academic or grammatical errors. The changes have been marked in red in the revised manuscript.

10) Writing: introduction section had highly similar contents repeated twice. Shorten it to be more succinct.

Reply: Thanks the reviewer for this suggestion. We have shorten the similar description in the introduction section. The changes have been marked in red in the revised manuscript.

Reviewer #3 (Remarks to the Author):

In this study, the authors used bioinformatic approaches to explore whether HCC-related RNA editing could be used to predict the risk, survival and immune responses of HCC tumors. A multi-step algorithm has been developed to identify RNA co-editing pairs, which then were used to stratify patients into different risk groups. Based on the risk stratification, patient survival and immune responses have been analysis and showed a significant correlation with RNA co-editing pairs and their related signaling pathways. Overall, the study provides new information of RNA editing in HCC biological changes and clinical outcomes in HCC tumors. There several questions regarding the data interpretation and analysis design:

Reply: Thanks very much for the overall positive comments.

1. In this analysis, if the expression of a gene is affected by an A-to-I RNA editing, that gene is considered to be a target gene for the given A-to-I RNA editing. It is used as a parameter to define RNA editing genes. Please clarify in the text that whether the affected expression is defined by increase or decrease of gene expression. It remains unclear to what degree the authors might find RNA co-editing pairs may lead to different directions of gene expression change and how this situation was processed during the analysis.

Reply: We thank the reviewer for this valuable suggestion. In our original paper, we compared gene expression levels between edited and non-edited samples to identify editing-associated genes (Wilcoxon-Mann-Whitney test, $FDR < 0.05$, Benjamini-Hochberg corrected), requiring a fold change of more than 2-fold, including editing with an increased or decreased gene expression change. As the reviewer 2 suggested, we performed the logistic regression to further filter the editing associated genes by adjusting age, gender, BMI, tumor stage and tumor grade in our revised manuscript ($FDR < 0.05$, Benjamini-Hochberg corrected). Finally, we identified 1,752,665 editing-gene interactions, of which 89.8% increase the gene expression while 10.2% have a decreased expression level. In our original paper, we identified A-to-I RNA co-editing pairs which significantly sharing the editing-associated genes by hypergeometric test (Benjamini-Hochberg correction, $FDR < 0.05$). However, in this process, we did not consider directions of gene expression change between two RNA editing sites. To avoid identifying RNA co-editing pairs which may associate with different directions of gene expression change, thus we restricted the shared editing-associated genes in the same direction. Finally, we obtained 86,822 interactions among 6,415 A-to-I RNA editing sites. We re-analyzed the results based on the reconstructed network, and found the subsequent results have remained largely unchanged. We have modified corresponding results in our revised manuscript.

Line 303-306:

Next, for each candidate RNA co-editing pairs, hypergeometric distribution was used to identify A-to-I RNA co-editing pairs that significantly shared the editing-associated genes (with consistent direction of gene expression change, Benjamini-Hochberg adjusted $FDR < 0.05$).

Line 99:

In total, we obtained 86,822 interactions among 6,415 A-to-I RNA editing sites (Figure 2A).

2. In Figure 2C, the authors found that 14.36% of RNA co-editing pairs resided on same chromosomes. In tumor cells, a whole chromosome and/or certain regions of chromosomes are amplified. Cancer cells are not diploid cells. Is there a relationship between the number of RNA co-editing pairs and the amplification of chromosomal numbers/regions, which may lead to the enrichment of co-editing events in the same chromosomal or the same gene regions? Does the genomic amplification or increased chromosomal number (chromosomal instability) affect the identification of co-editing pairs since these amplified genomic events may directly lead to an increased occurrence of co-editing pairs? The genomic data (chromosome information, amplification status) of these co-editing pairs need to be assessed and discussed to exclude these possibilities in Figure

2D and 2E (same chromosome/gene or different chromosome/gene are very vague).

Reply: Thank the reviewer for this constructive suggestion. We agree with the reviewer that amplified genomic events may directly lead to an increased occurrence of co-editing pairs, thus we assessed whether these RNA co-editing pairs prefer to be co-amplified in the genomic region (hypergeometric test, Benjamini-Hochberg correction, $FDR < 0.05$). It was revealed that 15.19% of RNA co-editing pairs showed to be co-amplified (13,184/86,822, Fig. S3A), and most of these co-amplified RNA co-editing pairs located in the same chromosome (85.54%, 11,277/13,184, Fig. S3B). Therefore, we agree with the reviewer that genomic amplification affects the identification of co-editing pairs and we discussed this in the revised manuscript (line 209-216).

On the other hand, we considered the possibility that amplified genomic events could directly contribute to an increased frequency of co-editing pairs. To investigate this, we evaluated whether these RNA co-editing pairs tended to be co-amplified within genomic regions (hypergeometric test, Benjamini-Hochberg correction, $FDR < 0.05$). We discovered that 15.19% of RNA co-editing pairs exhibited co-amplification (13,184/86,822, Fig. S3A), with the majority of these co-amplified RNA co-editing pairs being located on the same chromosome (85.54%, 11,277/13,184, Fig. S3B). These results suggest that genomic co-amplification can lead to an increased occurrence of co-editing pairs.

3. In addition, Figure 2E reported the gene location of RNA-editing. What are the targeting genes and what are the expression levels of these genes? These information could be reported to improve the clarity of analysis and also the interpretation of RNA-editing-associated changes.

Reply: Thank the reviewer for this suggestion. We have provided a supplementary table S1 to give the detailed information of RNA co-editing genes in same chromosomes. There are 12,537 RNA co-editing pairs located on same chromosomes, involving 921 genes, of which 116 genes were dysregulated between HCC tumor and adjacent normal samples (Supplementary table S, DEseq2, Benjamini-Hochberg corrected $FDR < 0.05$, with at least 2 fold changes). We have described this in the revised manuscript (line 106-110).

Among the 12,537 RNA co-edited pairs on the same chromosome, involving 921 genes, of which 2,457 co-editing pairs exhibited dysregulation of at least one genes between HCC and adjacent normal samples, which were involved in 116 dysregulated genes (DEseq2, Benjamini-Hochberg correction, false discovery rate (FDR) < 0.05 , with at least 2-fold changes, Supplementary table S1).

4. In Figure 7, the up-regulated genes and down-regulated genes were identified from normal, low and high-risk groups. It remains to be unclear whether there is a correlation between these genes (Figure 7C) and the genes identified from Figure 2E, and to what degree these genes in Figure 7 could be affected or associated with RNA-editing activities.

Reply: Thank the reviewer for this suggestion. We have checked the genes in figure 7C and genes in figure 2E, and found there are no shared genes between them. The expression levels of these seven genes are increased in the RNA co-editing HCC samples compared to those one without RNA co-editing (Figure S2, two-sided Wilcoxon-Mann-Whitney test, Benjamini-Hochberg corrected, $FDR < 0.05$). We have described this in the revised manuscript (line 176-178).

Globally, the expression levels of these seven genes were significantly elevated in HCC samples with RNA co-editing compared to those without (Figure S2, two-sided Wilcoxon-Mann-Whitney test, Benjamini-Hochberg adjusted $FDR < 0.05$).

5. In the line 200, the authors stated that "we identified eight critical genes regulated by five risk co-editing pairs". The data presented in Figure 7 is basically correlative analysis. These data do not provide causative relationship to support that the 8 genes are regulated by five risk co-editing. "Associated" is a more accurate statement rather than "regulated".

Reply: Thank the reviewer for this suggestion. Sorry for our inaccurate description and we have changed the term "regulated" to "associated" (line 242-243).

Thus RNA co-editing maybe associated with tumor immune microenvironment, and deserves further investigation.

6. In Figure 7D, multiple datasets were used to validate the prediction of genes identified from Figure 7C. However, can the authors validate their Figure 2D and Figure 2E results in multiple datasets? It is not clear whether repeatable or consistent RNA co-editing events can be identified from different datasets.

Reply: Thank the reviewer for this suggestion. As the discovery of RNA co-editing requires a sufficient number of HCC samples with both RNA sequencing and DNA mutation information, we have not found other suitable public datasets, thus here we did not validate the results of Figure 2D and Figure 2E in multiple datasets. We have admit this limitation in our revised manuscript (line 110-111).

These results require validation across multiple datasets.

7. The language could be improved by avoiding some small grammar mistakes: e.g. line 57, "Is RNA co-editing can be a clinical biomarker in HCC?"

Reply: We thank the reviewer for the suggestion. We apologize for the poor language of our manuscript. We have tried our best to improve the manuscript and have modified some confusing sentences, making them concise and easy to read. The changes have been marked in red in the revised manuscript, and we hope it could be acceptable for you.

e.g. "It raises the question: whether RNA co-editing could serve as a clinical biomarker in HCC?"

Version 1:

Reviewer comments:

Reviewer #1

(Remarks to the Author)

The authors have revised the manuscript and addressed the majority of comments.

The concept remains difficult to understand when considered against the known biology and likelihood that many editing events are stochastic and low frequency. This is reflected in the extensive clarifications requested in the revision. Despite this the data shows some statistical differences that associate with outcomes.

Reviewer #2

(Remarks to the Author)

Reviewer #3

(Remarks to the Author)

The revised manuscript has addressed the critiques and provided an adequate justification for publishing these findings. The reviewer has no further comments.
